# Genetic and Molecular Evidence of a Tetrapolar Mating System in the Edible Mushroom *Grifola frondosa*

**DOI:** 10.3390/jof9100959

**Published:** 2023-09-23

**Authors:** Shuang-Shuang Zhang, Xiao Li, Guo-Jie Li, Qi Huang, Jing-Hua Tian, Jun-Ling Wang, Ming Li, Shou-Mian Li

**Affiliations:** 1College of Horticulture, Hebei Agricultural University, Baoding 071001, China; 13722717690@163.com (S.-S.Z.); lixiao@hebau.edu.cn (X.L.); liguojie.imcas@foxmail.com (G.-J.L.); huang75661@163.com (Q.H.); yytjh@hebau.edu.cn (J.-H.T.); yyliming@hebau.edu.cn (M.L.); 2Hebei Key Laboratory of Vegetable Germplasm Innovation and Utilization, Baoding 071001, China; 3Collaborative Innovation Center of Vegetable Industry of Hebei Province, Baoding 071001, China; 4College of Life Science, Hebei Agricultural University, Baoding 071001, China; wangjunling2001@163.com

**Keywords:** *Grifola frondosa*, tetrapolar, mating-type locus, fine genomic map, SNP, indel

## Abstract

*Grifola frondosa* is a valuable edible fungus with high nutritional and medicinal values. The mating systems of fungi not only offer practical strategies for breeding, but also have far-reaching effects on genetic variability. *Grifola frondosa* has been considered as a sexual species with a tetrapolar mating system based on little experimental data. In the present study, one group of test crosses and six groups of three-round mating experiments from two parental strains were conducted to determine the mating system in *G. frondosa*. A chi-squared test of the results of the test-cross mating experiments indicated that they satisfied Mendelian segregation, while a series of three-round mating experiments showed that Mendelian segregation was not satisfied, implying a segregation distortion phenomenon in *G. frondosa*. A genomic map of the *G. frondosa* strain, y59, grown from an LMCZ basidiospore, with 40.54 Mb and 12 chromosomes, was generated using genome, transcriptome and Hi-C sequencing technology. Based on the genomic annotation of *G. frondosa*, the mating-type loci A and B were located on chromosomes 1 and 11, respectively. The mating-type locus A coded for the β-fg protein, HD1, HD2 and MIP, in that order. The mating-type locus B consisted of six pheromone receptors (PRs) and five pheromone precursors (PPs) in a crossed order. Moreover, both HD and PR loci may have only one sublocus that determines the mating type in *G. frondosa*. The nonsynonymous SNP and indel mutations between the A_1_B_1_ and A_2_B_2_ mating-type strains and the reference genome of y59 only occurred on genes *HD2* and *PR1*/*2*, preliminarily confirming that the mating type of the y59 strain was A_1_B_2_ and not A_1_B_1_. Based on the genetic evidence and the more reliable molecular evidence, the results reveal that the mating system of *G. frondosa* is tetrapolar. This study has important implications for the genetics and hybrid breeding of *G. frondosa*.

## 1. Introduction

*Grifola frondosa* is an edible fungus belonging to the phylum of Basidiomycota, the class of Agaricomycetes and the order of *Polyporales*. In Japan, its edible fruiting body is known as maitake. In China, *G. frondosa* is known as *HuiShuHua* (grey tree flower), perhaps because of its overlapping grey pileus. The broad-leaved forests of Asia, North America and Europe are habitable environments for the growth of the wood-decaying fungus *G. frondosa* [1]. Most frequently, *G. frondosa* is associated with the old trees of *Castanea* spp. and *Quercus* spp. [2]. With its delicious taste and various medicinal properties including antitumor [3], immune-function enhancing [4], hypoglycemic [5,6] and antiviral [7] activities and its ability to promote beneficial intestinal microbiota [8], there is an increasing demand for its fruiting body.

As a result of the success of the commercial cultivation of *G. frondosa*, breeding for strains with high-quality fruiting bodies and disease resistance are of increasing importance. For the successful production of desired strains, it is essential to clarify the mating system of this species. In general, three mating types were identified in fungi: homothallism (compatible with self), bipolar heterothallism and tetrapolar heterothallism [9]. Fungal mating systems are usually verified via three-round hybridizations [10,11,12,13,14], and the mating types of monokaryotic strains are accurately identified using the nuclear transfer test and OWE-SOJ (oak wood extract agar and squeezed orange juice agar) techniques [15,16,17]. These experiments are generally time-consuming and have relatively high rates of false detections; however, with the development of molecular biology technology, gene annotation and molecular markers can be used to determine mating systems. For example, by using molecular biology methods, the mating systems of the following macro-fungal species have been illuminated: *Agaricus bisporus* [18], *Flammulina filiformis* [19], *Ganoderma lucidum* [20], *Lentinula edodes* [21] and *Sparassis latifolia* [10].

In Agaricomycetes, the mating systems control the mating types of homokaryons, which are genetically haploid mycelia with a single nuclear type. When mating occurs between two homokaryotic mycelia with different mating-type nuclei, it results in the formation of a heterokaryotic mycelium with two nuclear types. Three kinds of genes are necessary for mating in Basidiomycota, including homeodomain transcription factors, pheromones and their cognate receptors. Tetrapolar mating-type systems depend on two unlinked loci called loci A and B [9,22]. The usual model of the HD locus (mating-type locus A) involves a divergently transcribed pair of homeodomain 1 (*HD1*) and homeodomain 2 (*HD2*) genes from different allelic subunits that form a functional group that regulates nuclear pairing, clamp cell formation and coordinated nuclear division [9,22]. The mating-type locus B (PR locus) consists of multiple pheromone precursors and pheromone receptor genes [23,24], promoting septal dissolution, nuclear migration towards the apical cell and clamp cell fusion [9,22]. In bipolar species, the two types of genes may be physically linked at a single mating-type locus, or only one of the two loci may determine the mating type [16,25].

Although *G. frondosa* has been considered a sexual species with a tetrapolar mating system, there is no genetic or molecular evidence to support this. In our study, protoplast monokaryotization and basidiospore-derived monokaryotic cultures of the two dikaryotic parent *G. frondosa* strains were generated, and mating experiments among the monokaryons were performed to ascertain the mating system, which was confirmed via resequencing. The genomic fine map at the chromosome level and the genetic structure of the mating-type loci A and B were analyzed for the first time. We provide genetic and molecular evidence of the mating system of *G. frondosa* that will be helpful for crossbreeding and for precise molecular breeding in the future.

## 2. Materials and Methods

### 2.1. Fungal Strains

The *G. frondosa* dikaryotic strains LMXY and LMCZ, which could develop mature fruiting bodies with basidiospores, were provided by the Edible Fungus Laboratory, Hebei Agricultural University (Baoding, China). Strains were preserved on potato dextrose agar medium (PDA: 200 g/L of potato infusion; 20 g/L of glucose; 20 g/L of agar) at 4 °C.

### 2.2. Methods for Acquisition of Monokaryotic Strains

A total of 77 (YS-1 to YS-77) monokaryotic strains were isolated from the *G. frondosa* dikaryotic strain, LMCZ, using protoplast monokaryotization according to a previously described strategy [17]. The mycelium of the LMCZ strain was cultured in potato dextrose broth (PDB, 200 g/L potato infusion; 20 g/L glucose) medium on a shaking table at 25 °C for about 5 d. The mycelium was collected and ground, 0.6 mol/L mannitol (Beijing Solarbio Science & Technology Co., Ltd., Beijing, China) was added as an osmotic pressure stabilizer and the suspension was centrifuged at 4000 r/min for 10 min. The mycelium was washed twice. Protoplasts were obtained via enzymolysis with 2% fungal lywallzyme (Institute of Microbiology, Guangdong Academy of Sciences, Guangzhou, China) at 30 °C for 3.5 h, filtered and centrifuged at 4000 r/min for 10 min, and 0.6 mol/mL mannitol was used as osmotic pressure stabilizer. Lastly, the protoplasts were spread on mannitol, yeast and glucose regeneration agar (MYG, 109.32 g/L mannitol, 4 g/L yeast, 4 g/L glucose, 20 g/L agar and 10 g/L maltose) and cultured at 25 °C for 7 d [17,26]. The newly germinated protoplasts were selected and transferred to PDA slants for maintenance culture. Strains without clamp connections were selected via microscopic observation, and the monokaryotic strains were stored for later use. To produce fruiting bodies for basidiospore isolation, the two *G. frondosa* dikaryotic strains, LMXY and LMCZ, were cultivated on PDA at 25 °C in darkness for 13–18 d and then transferred to sterile plastic bags containing growth substrate (chestnut sawdust, 36%; cottonseed hulls, 38%; bran, 15%; gypsum, 1%; sucrose, 1% and soil, 9%). The cultures were grown at 25 °C with 70% humidity for about 50 d in the dark, followed by cold stimulation at 18–22 °C with 70% humidity and light at 1200–2000 lx until primordia developed. The cultures were maintained at low temperature (18–22 °C, 85–95% humidity and 1200–1500 Lux) to allow for full fruiting body development. The fully mature pileus was removed and suspended in a 100 mL sterile triangular bottle for 24–48 h (Appendix A). About 5 mL of sterile water was added to the bottle, and spores were suspended in the liquid. The basidiospore suspension was diluted to 1 × 10^3^ spores/mL using sterile water, and 10 µL aliquots were evenly spread on 90 mm dishes containing PDA and incubated at 25 °C for about 7 d until the spores germinated. Two monokaryotic strains of *G. frondosa*, LMXY and LMCZ, were selected under an optical microscope (Eclipse Ci-L, Nikon Instruments Co., Ltd., Shanghai, China) and grown separately on PDA at 25 °C for 13–18 d.

### 2.3. Mating Experiments and Identification of Mating Types

Monokaryons for mating experiments were obtained via protoplast monokaryotization as described above. Two strains of monokaryotic hypha were inoculated with PDA in 90 mm Petri dishes 5 mm apart from the center and grown at 25 °C in darkness for 2 weeks. The mononuclear nature of the two strains was confirmed with self-crossing experiments and by checking the mycelia from the interacting regions of the two monokaryons for clamp connections using a visible light microscope [23]. The presence of multiple clamp connections was taken as confirmation of successful mating. Incompatibility reactions and absence of clamp connections signaled unsuccessful mating.

Test cross and three-round mating experiments were conducted to identify the mating types of the monokaryon strains from protoplast monokaryotization and sexual reproduction of the *G. frondosa* dikaryotic strain, LMCZ. Gridded charts were used to follow and verify the mating types of the monokaryons selected [27]. In the test-cross experiment, 77 monokaryotic protoplast strains were obtained via microscopic examination. The YS-7 strain from protoplast monokaryotization was selected randomly, putatively labeled as mating-type A_1_B_1_ and used as a test strain for test-cross determination. After YS-7 hybridization with the other 76 monokaryotic protoplast strains, the hyphae in the crosses were examined microscopically. Successful mating with this test strain allowed us to identify monokaryons of the opposite mating type (A_2_B_2_). The chi-squared test (χ^2^) was performed on the ratios of the two mating types among the 77 monokaryons. Basidiospores from the *G. frondose* strains LMXY and LMCZ were used to conduct three-round mating experiments, and intrastrain mating grids used to identify monokaryotic protoplast strains from parent strains are shown in Appendix A as described previously [10].

### 2.4. DNA/RNA Preparation

*Grifola frondosa* monokaryotic strain, y59, from basidiospores of the LMCZ strain was cultured on PDA at 25 °C in darkness for 10–15 d. Genomic DNA was isolated using a variation of the cetyltrimethyl ammonium bromide technique [28]. For transcriptome analyses, total RNA was isolated using the EZNA kit (Omega, Stamford, CT, USA).

### 2.5. Genome Sequencing and Assembly

The genome of *G. frondosa* strain, y59, was sequenced using the Pacific Sequel II method (sequencing depth of 100×) by Biomarker Technologies Co., Ltd. (Beijing, China). First, 39.53 Gb of sub-reads was obtained using PacBio’s circular consensus sequencing (CCS) program (https://github.com/PacificBiosciences/ccs (accessed on 10 February 2022)). To obtain more accurate sequencing data, the sub-reads were processed by means of Smartlink’s CCS (https://www.pacb.com/products-and-services/analytical-software/smrt-analysis/ (accessed on 10 February 2022)). Hifiasm [29] and Pilon [30] software were used to assemble the CCS reads and correct the assembled genome, respectively. After assembling the genome, its integrity was evaluated from the second-generation data return ratio and BUSCO (benchmarking universal single-copy orthologues) [31]. The genome sequence was deposited in the NCBI BioProject database (https://www.ncbi.nlm.nih.gov/bioproject (accessed on 6 January 2023)) under the accession number PRJNA918831.

Transcript RNAs were predicted using RNA Nano2000 assay kit (Thermo Scientific, Waltham, MA, USA) and the Qubit^®^ RNA assay kit on the Qubit^®^ 2.0 fluorometer (Life Technologies, Carlsbad, CA, USA). An RNA sample of *G. frondosa* strain, y59, was sequenced with the Illumina NovaSeq by Biomarker Technologies, Ltd. (Beijing, China). A total of 6 Gb of clean data was used for genome-assisted assembly.

### 2.6. Construction of Hi-C Libraries and Chromosomal Assembly

DNA isolation, library construction, sequencing and assembly were carried out by Biomarker Technologies, Ltd. (Beijing, China). The chromatin was digested with *HindIII* and ligated in situ after cross-linking with methyl aldehyde. DNA fragments were tagged with biotin, and those containing interactions were cyclized. Lastly, the DNA fragments were captured and purified using streptavidin-coated magnetic beads, and subjected to Illumina HiSeq sequencing. Hi-C sequencing libraries were constructed, and the concentration and insert size were determined using Qubit (2.0) (Thermo) and Agilent (2100) (Agilent) software. In addition, the effective concentration of the library was quantitated via q-PCR to ensure quality. High-quality samples were sequenced using Illumina Hiseq with read lengths of PE150. Clean reads were mapped to the *G. frondosa* genome using BWA [32]. Valid interaction pairs and invalid interaction pairs were identified using HiC-Pro (2.10) [33]. Data on the division, order and direction of scaffolds and contigs were used in Lachesis to assemble them into super scaffolds [34].

### 2.7. Genome Annotation

The combination of ab initio prediction, homology-based prediction and transcriptome-assisted prediction was used to identify the genes encoding proteins. For ab initio prediction, Genescan (1.0) [35], Augustus (2.4) [36], GlimmerHMM (3.0.4) [37], GeneID (1.4) [38] and SNAP (2006.07.28) [39] were used with default settings. The GeMoMa (1.3.1) module [40] was used for predicting homologous proteins.

RNA-seq data were mapped to the sponge gourd genome using Hisat2 (2.0.4) and StringTie (1.2.3) [41] for transcriptomics prediction. Unigene sequences were predicted from the transcriptome assemblies using TransDecoder (2.0) (Broad Institute, Cambridge, MA, USA). The results were integrated with EVM (1.1.1) [42] to identify the genes.

For gene functional annotation, the databases used included KOG [43], the Kyoto Encyclopedia of Genes and Genomes (KEGG) (http://www.genome.jp/kegg/ (accessed on 10 February 2022)) [44], Swiss-Prot (http://www.uniprot.org (accessed on 10 February 2022)) [45], TrEMBL (http://www.uniprot.org/ (accessed on 10 February 2022)) [45], gene ontology (GO) [46] and NR [47]. Other information on the *G. frondosa* genome was obtained from NCBI (https://www.ncbi.nlm.nih.gov/ (accessed on 10 February 2022)).

### 2.8. Structural Analyses for the Mating-Type Loci A and B

The structures of genes at the mating-type A and B gene loci were drawn using IBS (1.0.3) software, and GSDS (2.0) (http://gsds.gao-lab.org/ (accessed on 1 November 2022)) [48] was used to analyze the gene structures of *HDs*, *β-fg*, *MIP*, *PRs* and *PPs*. The conserved homeodomain of HDs at the mating-type A site was analyzed with CDD (http://www.ncbi.nlm.nih.gov/Structure/cdd/wrpsb.cgi (accessed on 15 April 2022)) [49]. The conserved nuclear localization signal (NLS) of HDs at the mating-type A site was analyzed using NLStradamus (http://www.moseslab.csb.utoronto.ca/NLStradamus/ (accessed on 15 April 2022)) [50] and PSORT II Prediction (http://psort.hgc.jp/form2.html (accessed on 15 April 2022)) [19]. TMHMM Server (http://www.cbs.dtu.dk/services/TMHMM-2.0/ (accessed on 15 April 2022)) [19] was used to analyze the transmembrane helix structure of pheromone receptors. Multiple amino acid sequence alignments of PRs were performed using ClustalW (https://www.genome.jp/tools-bin/clustalw (accessed on 15 April 2022)) [51].

### 2.9. Phylogenetic Analysis of HD1s, HD2s and PRs in Some Agaricomycetes Species

The amino acid sequences of HD1, HD2 and PR proteins of some Agaricomycetes including *Agaricus bisporus*, *Auricularia auricula-judae*, *Auricularia cornea*, *Heterobasidion irregulare*, *Hypsizygus marmoreus*, *Lentinula edodes*, *Lepista nuda*, *Pleurotus djamor*, *P. eryngii*, *P. ostreatus*, *P. tuoliensis*, *Sparassis latifolia* and *Volvariella volvacea* (a total of 37 amino acid sequences of HD1 protein, a total of 53 amino acid sequences of HD2 protein and a total of 42 amino acid sequences of PR protein) were obtained from GenBank (https://www.ncbi.nlm.nih.gov/ (accessed on 25 October 2022)). The amino acid sequence alignments and phylogenetic tree construction with maximum likelihood (ML) between the *G. frondosa* HD1s, HD2s and PRs and the downloaded HD1s, HD2s and PRs were conducted using MEGA X [52]. The optimal evolutionary model was determined as LG + G using the find-best-protein-model option in MEGA-X. The ML tree was constructed based on the nearest-neighbor-interchange method and the use-all-sites-for-gaps option in MEGA-X. The consistency of the phylogenetic estimate was established using the ultra-fast bootstrap method with 1000 replicates.

### 2.10. Allelic Difference Analyses of Mating-Type Loci A and B

A total of 23 A_1_B_1_ and 22 A_2_B_2_ strains from the protoplast monokaryotization of the LMCZ strain were selected for next-generation sequencing (hyphal sample preparation for sequencing was the same as for 2.4 DNA/RNA preparation). The sequencing reads obtained from them were relocated to the whole genome of monokaryotic strain, y59, in this study for subsequent sequence variation analysis. BWA [53] software was used to compare the short sequences obtained via second-generation high-throughput sequencing with the reference genome. Single nucleotide polymorphisms (SNPs) and small insertions and deletions (indels) were detected using the GATK [54] software. The mutation sites of SNPs and indels were obtained with Samtools (1.9) [55], and the mutated genes were compared with NR [47], KEGG [44] and other functional databases by BLAST [56] to obtain the annotations of these genes and to analyze their functions. Lastly, the differential sequences of mating-type loci A and B alleles containing the SNPs and indels were obtained.

Re-sequencing strains include the following:

A_1_B_1_: YS-4, 7, 10, 12, 13, 15, 16, 18, 20, 23, 26, 28, 33, 34, 40, 46, 55, 60, 61, 62, 66, 67, 75;

A_2_B_2_: YS-5, 8, 11, 17, 19, 21, 24, 25, 27, 29, 42, 44, 45, 51, 63, 65, 68, 69, 70, 72, 73, 76.

## 3. Results

### 3.1. Protoplasts and Basidiospores of Monokaryotic Cultures

The fruiting bodies of *G. frondosa* were produced on a substrate of chestnut sawdust. The fruiting bodies of the dikaryotic strains LMXY and LMCZ had coralline branching with light and dark taupe pileus, respectively (Figure 1A,B), and their basidiospores were observed (Figure 1C). The protoplasts were obtained after the young hyphae were treated with wall-degrading enzymes (Figure 1D). All the basidiospores and protoplasts were observed to readily germinate, suggesting that they had high viability. Finally, 246 (Q-1 to Q-246) and 123 (y-1 to y-123) monokaryotic strains were obtained from the basidiospores of the LMXY and LMCZ strains, respectively. A total of 77 (YS-1 to YS-77) monokaryotic protoplast strains were obtained via the protoplast monokaryotization of the LMCZ strain.

### 3.2. Mating Experiments and Mating Type Identification

In the test-cross experiments, successful crossed and positive mating demonstrated numerous clamp connections (Figure 1E). Out of the 77 monokaryons, we selected YS-7 as the test strain. Although the ratio of the mating types among the monokaryons from the parental strain in the test crosses was not the expected 1:1 ratio, the χ^2^ test indicated little distortion among the tested strains, confirming the accuracy of the results (Table 1 and Appendix A).

In the three-round mating experiments, the mating types of 123 basidiospore strains from the LMCZ strain were determined, and the YS-7 strain was still selected as the test strain. The mating experiment results were consistent with only two mating types, A_1_B_1_ and A_2_B_2_, but there were more A_1_B_1_ strains than A_2_B_2_ strains (Table 2 and Appendix A). The same experiment was conducted with YS-5 and YS-11, both A_2_B_2_ type strains, as indicated by the test strain. Not surprisingly, there were no A_1_B_2_ or A_2_B_1_ strains found among the 123 basidiospore strains (Table 2 and Appendix A). Another *G. frondosa* dikaryotic strain, LMXY, was chosen to prepare basidiospore strains and conduct the same mating experiment. Although the numbers of A_2_B_1_ and A_1_B_2_ strains were far smaller than the numbers of A_1_B_1_ and A_2_B_2_ strains, all four mating types were present among the strains (Table 2 and Appendix A).

### 3.3. Fine Genomic Map of Grifola frondosa Strains y59 from LMCZ 

To investigate the mating-type loci and the mating types of *G. frondosa*, a genomic map of the monokaryotic strain, y59, was constructed using genome, transcriptome and Hi-C sequencing technology. The complete genome of *G. frondosa* was 40.54 Mb with a contig N50 of 3,000,501 bp. The average GC content in the corrected genome was 49.57%, and the sequencing depth was 975.06×. A total of 13,437 protein-coding genes were identified, and 10,999 genes were annotated using BLAST comparison with the KOG and KEGG functional databases. After the Hi-C assembly was conducted, 38.1 Mb was sequenced and assigned to 12 chromosomes whose proportion was 93.99% (Appendix A). To the best of our knowledge, this is the first chromosome-level genome assembly for *G. frondosa*, and a detailed analysis of the genome will be presented in a future paper.

### 3.4. Mating-Type Loci in G. frondosa

The fine genomic map of the *G. frondosa* monokaryotic strain, y59, revealed that the two mating-type loci, A and B, were located on chromosomes 1 and 11, respectively (Figure 2). In the tetrapolar heterothallic species of Basidiomycota, the mating-type loci A and B are found at two unlinked positions, controlling the formation and stability of the heterokaryon during mating [57]. Clearly, *G. frondosa* is a tetrapolar species.

#### 3.4.1. Mating-Type Locus A

The mating-type locus A contained one β-fg protein (EVM01G010490.1), two HD proteins (EVM01G010500.1 and EVM01G010510.1) and one MIP protein (EVM01G010520.1) (Figure 2). There was only one HD2 with two nuclear localization signal sites and one homeodomain in *G. frondosa* (Figure 2 and Appendix A), and it was grouped with the HD2s of *A. auricula-judae* in the phylogenetic analysis (Figure 3B and Appendix A). On the left of HD2 with a ‘head-to-head’ gene coding direction, there was a protein with one C-terminal domain of homeodomain 1 and one nuclear localization signal site without a DNA-binding domain (Figure 2 and Appendix A). According to the phylogenetic analysis of the HD1s in some Agaricomycetes, the protein with the C-terminal domain of homeodomain 1 in *G. frondosa* should be grouped with other HD1s (Figure 3A), and we surmised that this protein was HD1. HD2 was adjacent to MIP, HD1 was adjacent to β-fg protein and HD1 and MIP shared the same gene coding direction (Figure 2).

At the gene structure of the mating-type locus A, three of the four genes shared four exons and three introns, and only the *β-fg* gene had six exons and five introns (Appendix A).

#### 3.4.2. Mating-Type Locus B

In the genome of *G. frondosa*, the mating-type locus B consists of six pheromone receptor genes (*PR1*–*PR6*) (EVM11G001920.1, EVM11G001930.1, EVM11G001950.1, EVM11G001970.1, EVM11G001980.1 and EVM11G00200.1) and five pheromone precursor genes (*PP1*–*PP5*) (EVM11G001880.1, EVM11G001940.1, EVM11G001960.1, EVM11G001990.1 and EVM11G002090.1). The total length of mating-type locus B, calculated from the first gene (*PP1*) to the last gene (*PP5*), was 47,772 bp (Figure 2). There was a single PR sublocus at a distance of 10.8 kb from *PP1* and at a distance of 12.5 kb from *PP5*. The six *PRs* shared five to seven exons, and *PR2* had the greatest length at 4589 bp, but the other five *PRs* were also quite long (1778 bp–2944 bp) (Appendix A). Pheromone receptors are G-protein-coupled receptors that are bound to the cell membrane by a seven transmembrane (7-TM)-spanning domain, with the N-terminus outside and the C-terminal end inside [58,59]. *Grifola frondosa* not only had four PRs (PR1, PR3, PR5 and PR6) with seven helical transmembrane-spanning structures, but also had PR2 with five TM helices and PR4 with four TM helices based on the phylogenetic analysis, amino acid sequence alignment and TMHMM Server results (Figure 3C and Appendix A).

As the pheromone precursor gene and protein sequences are not well conserved, they cannot be obtained using the BLAST sequence alignment, but only by searching for the conserved motifs of the pheromone precursor amino acid sequences, such as CAAX (C, cysteine; A, aliphatic amino acids; X, alanine, serine, methionine, glutamic acid, cysteine, etc.), EA (E, glutamic acid; A, alanine) and AF (F, phenylalanine) base sequences [59,60]. Five suspected pheromone precursor genes were found in the *G. frondosa* strain, LMCZ, according to the protein sequences (Appendix A). The five *PPs* shared two to five exons, but each had a sequence length that was different from the others—*PP2* was the longest with 1701 bp, while *PP1* was the shortest at 221 bp (Appendix A). All the *PRs* and *PPs* were distributed crossly and dispersedly (Figure 2 and Appendix A).

### 3.5. SNP and Indel Mutations in Mating-Type Locus A Alleles

For a general analysis of the mating types, the whole genomes of 23 A_1_B_1_ and 22 A_2_B_2_ monokaryotic protoplast strains from the *G. frondosa* strain, LMCZ, were re-sequenced with y59 as the reference genome. The average mapping rate between the samples and the reference genome was 95.33%, and the average coverage depth was 87×. Most SNPs and small indel mutations were distributed on the mating loci, but the nonsynonymous SNPs and small indel mutations also occurred on the *HD2*, *PR1* and *PR2* genes; only *PR2* belonged to the PR sublocus (Figure 2 and Appendix A). We preliminarily inferred that sublocus allelic variations exist on *HD2* and *PR2*, which determines the mating types in *G. frondosa*, and there may be four different mating types.

At the mating-type locus A, the SNP mutation results show that only synonymous SNP sites existed among the 23 A_1_B_1_ and 22 A_2_B_2_ monokaryotic protoplast strains compared with y59 (Appendix A). For small indel mutations, there were also synonymous intron, upstream and downstream mutations at the mating-type locus A in 22 A_2_B_2_ monokaryotic protoplast strains, similar to the SNP results (Figure 4). There were eight nonsynonymous indel mutations at the *HD2* locus of 22 A_2_B_2_ monokaryotic protoplast strains, including seven frame shifts (insertion or deletion multiples of non-3) and one stop codon insertion (Figure 4). Other mating-type locus A genes, such as *MIP*, *HD1* and *β-fg*, were only found to have synonymous indel mutations at the upstream and downstream regions, with no effect on the protein function (Figure 4). These results imply that the mating-type locus A of y59 was the same as in the 23 A_1_B_1_ monokaryotic protoplast strains, but differed from that of the 22 A_2_B_2_ monokaryotic protoplast strains.

### 3.6. SNP and Indel Mutations in Mating-Type Locus B Alleles

Whole genome sequencing disclosed many more SNPs and indel mutations at the mating-type locus B compared with locus A. More importantly, only the pheromone receptor genes, *PR1* and *PR2*, of 23 A_1_B_1_ monokaryotic protoplast strains had nonsynonymous SNPs and indel mutations compared with y59 (Figure 5 and Appendix A), suggesting that y59 shared the same B mating type with 22 A_2_B_2_ monokaryotic protoplast strains, but differed from the 23 A_1_B_1_ monokaryotic protoplast strains.

With regard to the nonsynonymous SNP mutations of *PR1*, we found that certain types of mutations predominated; the 3′-UTR region, nonsynonymous coding variations, splicing region variation, stop codon gain, start codon gain, stop codon lost and 5′-UTR regions of *PR1* occurred the most (18 times) (Figure 5A). For *PR2*, nonsynonymous coding variation mutations were the most frequent, occurring 57 times (Figure 5A). The indel mutation results of *PR1* and *PR2* show that there were far fewer nonsynonymous indel mutations than nonsynonymous SNP mutations. There were five nonsynonymous indel mutations in *PR1* (one splice site and four 5′-UTR regions) and four nonsynonymous indel mutations in *PR2* (two frame shift, one codon deletion and one stop codon gained) (Figure 5B). Taken together, these observations suggest that the mating type of the y59 monokaryotic strain was A_1_B_2_ and not A_1_B_1_.

## 4. Discussion

In Agaricomycetes, the mating types of homokaryons, genetically haploid mycelia with a single nuclear type, are controlled by mating systems. When a homokaryotic mycelium mates with one of a different nuclear mating type, a heterokaryotic mycelium with two nuclear types is formed. Three mating systems can be attributed to the group: homothallic, bipolar heterothallic and tetrapolar heterothallic mating systems [61]. A mating system confirmation is necessary to identify new varieties for mushroom crossbreeding. As a valuable edible fungus, *G. frondosa* has been considered a sexual species with a tetrapolar mating mode without evidence to support that claim. Here, we performed both mating experiments with protoplasts and monokaryotic cultures derived from basidiospores. The results of the analyses on the mating-type genes based on a chromosome-scale genome assembly and SNP/indel sequencing confirmed that *G. frondosa* is a tetrapolar fungus and disclosed the gene structural features of the mating-type loci in *G. frondosa*.

Mating methods are commonly used to identify the mating type [62,63]. In this study, three-round mating experiments were conducted to determine the mating system and identify the different mating-type strains from the basidiospores of two strains, each with three different test strains. While the observed mating type ratio of the monokaryotic basidiospore strains, LMXY and LMCZ, was not 1:1:1:1, we discovered only two kinds of mating types in the LMCZ strain (Table 2). However, the chi-squared test of the test-cross mating experiments with the monokaryotic protoplasts from the LMCZ strain satisfied Mendel’s rule of segregation (Table 1). Segregation distortion was found in many edible fungi species, such as *Agrocybe salicicola* [64], *L. edodes* [65] and *Pholiota adiposa* [66], and even among some plants [67,68]. The distorted segregation of the mating types of Basidiomycetes may be the result of a lower germination capacity or growth rate of basidiospores [64], the presence of heterozygotes or even the existence of pseudo-clamp connections. In *Gloeostereum incarnatum*, 14 segregation distortion markers were detected in the genetic linkage map, which improved the efficiency of genetic breeding and new variety selection [69], and suggested the existence of a tetrapolar mating system in *G. frondosa*.

At the molecular level, the tetrapolar mating system is based on two unlinked mating-type loci, A and B, which are located on different chromosomes. In *G. frondosa*, the unlinked mating-type loci, A and B, were on chromosomes 1 and 11 (Figure 2); this fact, together with the results of the mating experiments, confirm that the mating system of *G. frondosa* is tetrapolar.

In most edible fungi, the mating-type locus A usually contains two HDs: HD1 and HD2. When an HD1 protein heterodimerizes with an HD2 partner encoded by different allelic versions of the same mating-type A group, it will induce the expression of a transcription factor that is specific for diploid cells [57,70,71]. Moreover, Agaricomycetes may have one or more HD subloci. *Laccria bicolor* [72], *Lentinula edodes* [73] and *Pleurotus djamor* [74] only have one HD sublocus, *F. filiformis* has two HD subloci [19] and *Coprinus cinereus* contains three HD subloci [72,75]. In *G. frondosa*, only one HD sublocus was found, including one HD2 and a protein containing a C-terminal domain of homeodomain 1 that is different from the homeodomain 1 of HD1. The phylogenetic analyses of HD1s in the Agaricomycetes showed that the protein with the C-terminal domain of homeodomain 1 in *G. frondosa* was indeed grouped with the HD1s of other species (Figure 3A). Moreover, Kües et al. found that only the homeodomain motif of HD2, as a potential DNA binding domain, could be fused to an essential C-terminal region of the HD1 protein to form an active HD1-HD2 protein complex [76]. In other words, the protein with the C-terminal domain also possessed the function of the HD1 protein and might not have a relationship with the formation and function of HD1-HD2 heterodimerization in *G. frondosa*. To understand the regulation of the HD pathway in *G. frondosa*, it will be necessary to investigate the dimerization of HD1 and HD2 by conducting analyses of the A locus of different monokaryotic strains of LMCZ as well as to investigate the formation process of dimerization in a future study.

The HD locus is always flanked by the *MIP* gene, and a gene that encodes *β-fg* most of the time at the mating-type locus A is commonly observed in other basidiomycetes [57]. The *MIP* and *β-fg* genes are found in *G. frondosa*, and they were located on both sides of the *HD2* and *HD1* genes, respectively, meaning that the mating-type A region of *G. frondosa* displayed extensive synteny across the Polyporales (Figure 2) [57]. The gene sequence of the mating-type locus A of *G. frondosa* was the same as that of *G. lucidum* and *Lentinus tigrinus* (Appendix A) [10,19,20,21,72,73,74,75,77,78].

The mating-type locus B codes for specific peptide pheromones and their receptors, which control nuclear migration and the fusion of clamp connections [70]. Each pheromone can bind to a receptor encoded by different allelic versions of the same group of mating B genes, thus activating a downstream signaling pathway to control nuclear migration and the fusion of clamp connections [70]. There is only one PR sublocus in *G. frondosa* (this study) and *S. latifolia* [79], while other basidiomycetes can have more than one, such as *F. filiformis* [19] and *L. edodes* [58], which have two PR subloci. The fungal pheromone receptors usually contain seven transmembrane (7-TM) domains, extracellular N-termini and intracellular C-termini [80]. The number of PRs and their TM domains varies among different species. For example, *Schizophyllum commune* owns four PRs with 7 TM domains [81], *Stropharia rugosoannulata* contains five PRs with 3–7 TM domains [82], *Pleurotus pulmonarius* has eight PRs with 3–8 TM domains [83] and *G. frondosa* has six PRs with 4–7 TM domains.

Most studies on mating-type A and B loci were carried out via genome resequencing and the alignment of different mating-type strains [18,60,79]. In this study, we carried out the next-generation sequencing of the monokaryotic basidiospore strains, A_1_B_1_ and A_2_B_2_, and of the *G. frondosa* strain, LMCZ. The alignment of these data with the genome of y59 showed that nonsynonymous SNPs and small indel mutations only occurred at the genes *HD2*, *PR1* and *PR2*, implying that these three subloci generated allelic variations among different mating-type strains in *G. frondosa*. In *F. filiformis*, the mating-type locus A included an incomplete HD-a and a complete HD-b; the mating-type B allele of the L11 strain (A_1_B_1_) and the KACC42780 strain (A_3_B_3_) contained the same PR-a sublocus, but contained a different PR-b sublocus, meaning that both the HD and PR subloci could recombine and generate new mating types [18,84]. Studies on *Coprinopsis cinerea* have reported that the HD locus is multi-allelic and composed of two subloci [85]. In the SP3 and SP30 strains of *L. edodes*, however, the HD loci both generated a new *HD* gene with only one quarter of the size of the *HD2* gene, and the PR loci contained an incomplete PR sublocus with high homology to the *rcb* and *phb* genes, implying that there were several pairs of HD and PR loci within the individual genomes in *L. edodes* [21]. Thus, studying the mating-type loci could enhance the efficiency of future breeding programs and reduce strain degeneration for better cultivar maintenance.

In this study, the SNP and indel mutations in two mating-type strains compared with y59 preliminarily allowed us to conclude that the mating type of the y59 strain was A_1_B_2_ and not A_1_B_1_. The most important standard of mating events was clamp connection formation, and false positive results could occur during mating. The cytodifferentiation terminated here in the common-B heterokaryon (A ≠ B =) and the resultant appendage was a pseudo-clamp connection [86], as described in *S. latifolia* [10] and *P. citrinopileatus* [87]. During our experiments, it was found that the colony morphology at the intersection of two hyphal clumps from the same group of strains was different at different times. Moreover, a microscopic examination revealed that some hyphae at the intersection of two colonies with obvious antagonistic lines still showed clamp connections, implying that they might be pseudo-clamp connections and thus difficult to distinguish via microscopic examination alone. Nuclear DNA was stained with DAPI, and the FAA fixative method and Giemsa dye method [10,88] were used to stain the hyphae of *G. frondosa*. Hyphal septa could not be seen under the microscope, indicating that the mating experiments could not precisely distinguish *G. frondosa* strains with different mating types. In the future, the subloci of mating-type A and B loci will be further studied. Meanwhile, the development of mating gene molecular markers in *G. frondosa* is urgent.

In this study, we generated a 38.1 Mb genome fine map of *G. frondosa* strain, y59, which was grown from an LMCZ basidiospore, and confirmed via mating experiments and a mating-type loci analysis that *G. frondosa* is a tetrapolar species, and that the HD and PR loci may only consist of a single sublocus; we also determined the mating type in *G. frondosa*. These results demonstrate the importance and reliability of a mating-type loci analysis. We sincerely hope that this work will lay a theoretical foundation for the development of mating-type molecular markers, greatly improve the efficiency of hybridization and provide a theoretical basis for the precise molecular breeding of *G. frondosa*.

## Figures and Tables

**Figure 1 jof-09-00959-f001:**
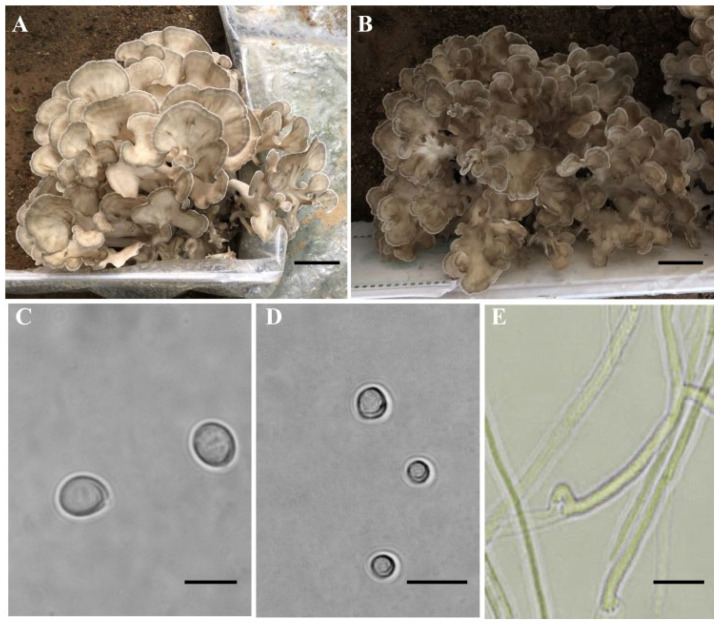
Mature fruiting bodies, basidiospores, protoplasts and clamp connections of *Grifola frondosa*. (**A**) Fruiting body of *G. frondose* dikaryotic strain LMXY, (**B**) fruiting body of *G. frondosa* dikaryotic strain LMCZ, (**C**) basidiospores, (**D**) protoplasts and (**E**) clamp connections. Bars: A, B = 5 cm; C, D = 10 μm.

**Figure 2 jof-09-00959-f002:**
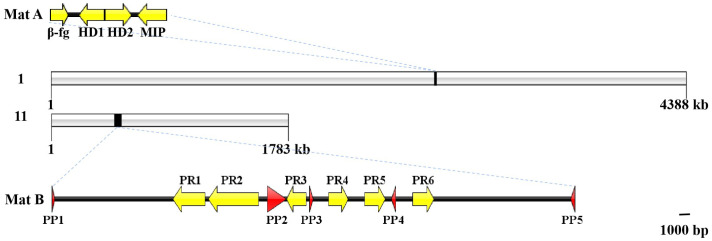
Chromosomal location of mating-type loci A and B in *Grifola frondosa*. Note: 1 and 11 represent chromosomes 1 and 11. The scale label on the bottom right refers to mating-type loci A and B.

**Figure 3 jof-09-00959-f003:**
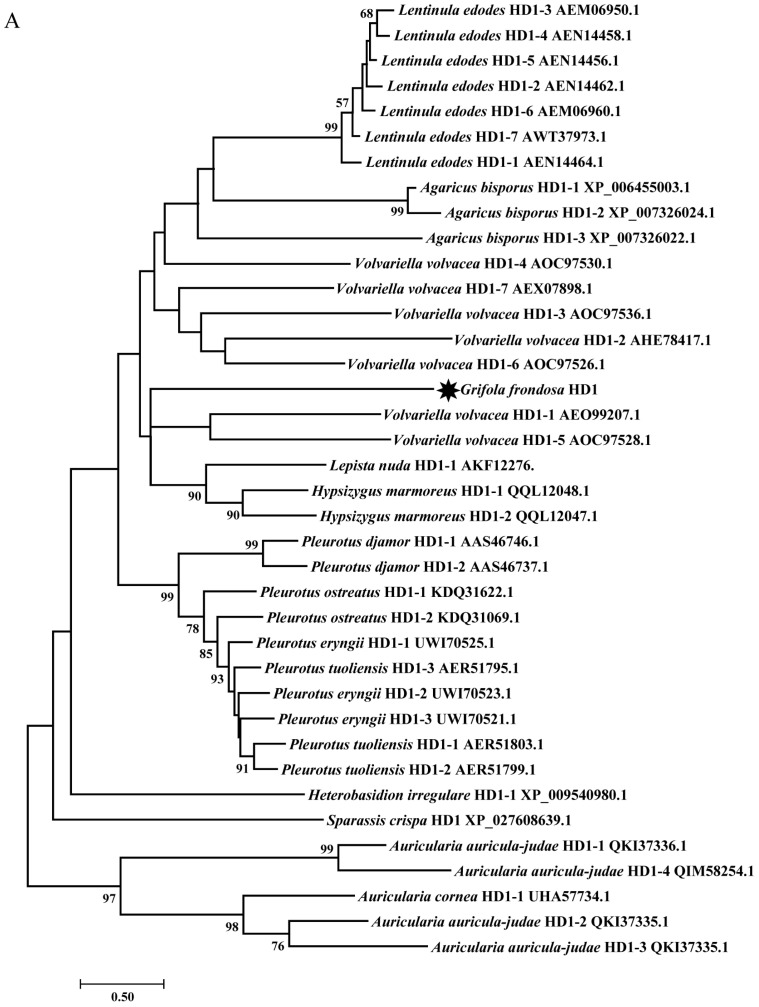
Maximum likelihood phylogenetic trees of (**A**) homeodomain encoding protein type 1 (HD1), (**B**) homeodomain encoding protein type 2 (HD2) and (**C**) pheromone receptor (PR) of some Agaricomycetes species. The black stars represent the proteins in *G. frondosa*.

**Figure 4 jof-09-00959-f004:**
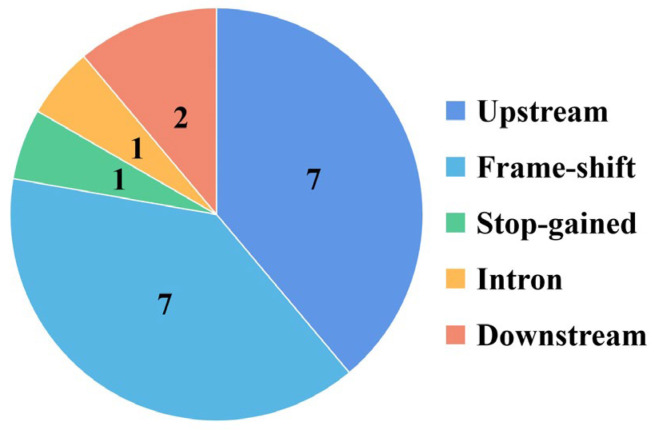
Indel mutations at mating-type locus A represented by A_2_B_2_ monokaryotic protoplast strains and reference genome, y59.

**Figure 5 jof-09-00959-f005:**
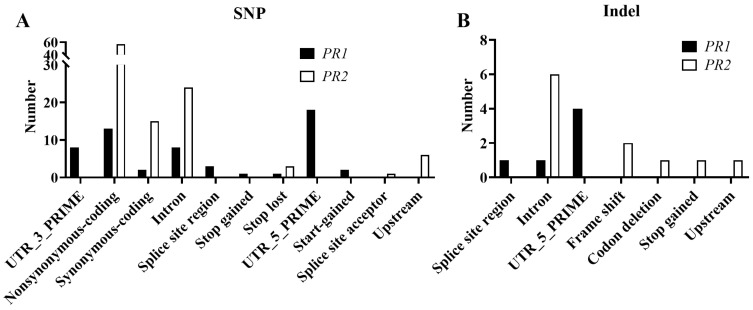
SNP (**A**) and indel mutations (**B**) in the *PR*1 and *PR*2 genes of mating-type A_1_B_1_ monokaryotic protoplast strains and the reference genome, y59.

**Table 1 jof-09-00959-t001:** Ratio of two mating types of YSs derived from *Grifola frondosa* strain, LMCZ.

Number of Monokaryotic Protoplasts	A_1_B_1_:A_2_B_2_	χ^2^
77	45:32	2.195

Note: χ^2^_0.05,1_ = 3.84.

**Table 2 jof-09-00959-t002:** Ratio of four mating types of basidiospores derived from parental strains LMXY and LMCZ.

*Grifola frondosa* Strains	Test Strains	A_1_B_1_:A_2_B_2_:A_2_B_1_:A_1_B_2_	χ^2^
LMXY	Q34	97:111:13:25	120.24
	Q39		
	Q4		
LMCZ	YS-7	83:40:0:0	153.07
	YS-5		
	YS-11		

Note: χ^2^_0.05,3_ = 7.81.

## Data Availability

The data presented in this study are available in Appendix A.

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
