# Peer review of "Genetic and Molecular Evidence of a Tetrapolar Mating System in the Edible Mushroom Grifola frondosa"

_jof, 2023, doi:10.3390/jof9100959_

Round 1

Reviewer 1 Report (Previous Reviewer 1)

Dear Authors 

The research publication "Genetic and Molecular Evidence for a Tetrapolar Mating System in the Edible Mushroom, Grifola frondosa" by authors Zhang et al. "investigated the G. frondosa [maitake] an edible species to hybridization." The following are the author's comments:

The authors clearly described the reproductive strategy and the difficulties encountered during hybridization. In the introduction, the authors also described the necessary stages for hybridization. 

Materials and methods

In section 2.2, authors must describe the protoplast isolation protocols, enzymes, and osmatic solution product details and provide relevant citations. The regeneration medium must include MgSO4. MgSO4 was the most effective agent for the discharge of protoplasts and can increase the regeneration rate. Check the space in the description of MYG media composition 

Mating types and selection of monokaryones, molecular identification, and genome sequencing should be expressed using the flow conversation method, which is more engaging to the readers.

Authors should replace Figures 1 C, D, and E with high-resolution images if practicable.

The BioProject number [PRJNA918831] was not found in the NCBI database. Authors should provide online verification information if genome sequencing and assembly cannot be accessed online. 

Author Response

Responses to Reviewer 1 Comments

The research publication "Genetic and Molecular Evidence for a Tetrapolar Mating System in the Edible Mushroom, Grifola frondosa" by authors Zhang et al. "investigated the G. frondosa [maitake] an edible species to hybridization." The following are the author's comments:

The authors clearly described the reproductive strategy and the difficulties encountered during hybridization. In the introduction, the authors also described the necessary stages for hybridization.

Response: Thanks for the positive comments. We have revised the ms carefully according to your suggestions.

Materials and methods

In section 2.2, authors must describe the protoplast isolation protocols, enzymes, and osmatic solution product details and provide relevant citations. The regeneration medium must include MgSO4. MgSO4 was the most effective agent for the discharge of protoplasts and can increase the regeneration rate. Check the space in the description of MYG media composition.

Response: Thanks for the suggestion. Protoplast isolation protocols, enzymes, and osmotic solution product details have been added (lines 102-111).

According to published data, mannitol works better than MgSO4 as an osmotic pressure stabilizer [26]. (Xue, P.H.; Gong, Z.; Xie, K.P.; Ma, H.; Zhao, H.; Cao, W.W. Study on the preparation and regeneration conditions of the protoplast isolation protocols of Grifola frondosa. Edible Fungi 2004, 01, 13-15. (in Chinese).

Mating types and selection of monokaryones, molecular identification, and genome sequencing should be expressed using the flow conversation method, which is more engaging to the readers.

Response: We wrote the paper in the order of the experiments. When we used molecular identification (SNP and indel mutation analysis), the genome was needed as reference, and therefore, we performed the genome sequencing before the molecular identification to make it easier to read.

Authors should replace Figures 1 C, D, and E with high-resolution images if practicable.

Response: Agreed and replaced.

The BioProject number [PRJNA918831] was not found in the NCBI database. Authors should provide online verification information if genome sequencing and assembly cannot be accessed online.

Response: As the genome of G. frondosa will not be released until later, we provide the reviewer link:

https://dataview.ncbi.nlm.nih.gov/object/PRJNA918831?reviewer=i6br5d7bi9iqscl2sfgigojclg

Reviewer 2 Report (New Reviewer)

In the manuscript ” Genetic and Molecular Evidence for a Tetrapolar Mating System in the Edible Mushroom, Grifola frondose”  the tetrapolar mating-type system has been characterized by mating interactions between monokaryotic strains originating from individual basidiospores or from protoplasting a dikaryon. The material originates from two dikaryotic strains LMXY and LMCZ that produce fruiting bodies and basidiospores. The main part of the work is done with the strain LMCZ: the whole genome sequencing of strain y59, and production of 23 protoplastic strains named  A1B1 and 22 protoplastic strains named A2B2, that all were sequenced. The research includes a huge amount of work and the molecular biology of the work seems to be well done. However, the presentation leaves lot to be corrected and requires major revision.

The list of revisions needed, there are probably more

Material and methods

line 93  The G. frondosa dikaryotic  bipolar strains LMXY and LMCZ, it is said at the end of Introduction that G. frondose is assumed to be tetrapolar.

line 137 The same mistake G. frondosa dikaryotic bipolar strain LMCZ What is meant with bipolar!!

line 154 Grifola frondosa monokaryotic strain y59 from basidiospores of strain LMCZ/ from a basidiospore of LMCZ

line166 Transfer RNAs were/ Transcript RNAs probably

Results

line 263 In the three-round mating experiments (reference? The method It is very well and clearly explained in refrence10, but not here)

Table 1 is all right, but for Table 2 it is necessary to do a three-round mating experiments with monokaryons grown from individual basidiospores collected from LMCZ fruiting bodies. Then there would be similar distribution of A and B mating -type as from LMXY. Note in Abstract and at the end of the manuscriptit is said Y59 grown from a LMCZ basidospore has the genotype A1B2

I do not see any sense in the three-round matings between YS-7, YS-5 and YS-11 (protoplastic strain from a dikaryon with mating-types A1B1 and A2B2) and 123 monokaryotic strains grown from single basidiospores of LMCZ . The123 spores should be used for the three-round experiment suggested above.

line 287 3.3. Fine genomic map of Grifola frondosa strain LMCZ Y59

line 299 The fine genomic map of the G. frondosa monokaryotic strain, y59, revealed that the two mating-type loci A and B, were located on chromosomes, 1 and 11. What are the bases of the A and B locus recognization ???

chapter 3.4.1 has to be rewritten, for instance: “There was only one HD2 with two nuclear localization signal sites and one homeodomain in G. frondosa (Figure 2, Supplementary Figure S3B) that were grouped to the HD2s of A. auricula-judae in phylogenetic analysis/ divide in two sentences

For B-receptor and pheromone working in mating the best candidates would be PR3 (sevenTMs) and PP3 nice carboxcyl end CVIAA

The chapters 3.5. and 3.6. dealing with SNP and indels in A and B are poorly written line 363 “We preliminarily inferred that sublocus allelic variation exists on HD2 and PR2 which determines the mating types in G. frondosa, and there may be four different mating types in G. frondosa.” PR2 with only 5 proper TMs?? See the note above

Fig.5 text  A  protoplastic strains A1B1? and B Y59 ?

Discussion

Segregation distortion? Difficulty to find equal number of the four mating-types???? Is that what the authors mean?

line 449-450 …to investigate the dimerization of HD1 and HD2 in  future by analyzes of A locus of different monokaryotic strains of LMCZ

line 457  G. lingzhi/ in S11 Ganoderma lucidum

line 467 The number of PRs and their TM domains… The authors should familiarize themselves with the publication What role might non-mating receptors play in Schizophyllum commune. J.of Fungi 7,399 (2021)doi.org/10.33390/jof7050399

lines 488-503, what is the purpose of this chapter, could be removed.

Supplementary material

text of S4 A ja B impossible to read.

The English of the text is relatively good, but several sentences could be clearer and thus improve the manuscript

Author Response

Responses to Reviewer 2 Comments

In the manuscript ”Genetic and Molecular Evidence for a Tetrapolar Mating System in the Edible Mushroom, Grifola frondose” the tetrapolar mating-type system has been characterized by mating interactions between monokaryotic strains originating from individual basidiospores or from protoplasting a dikaryon. The material originates from two dikaryotic strains LMXY and LMCZ that produce fruiting bodies and basidiospores. The main part of the work is done with the strain LMCZ: the whole genome sequencing of strain y59, and production of 23 protoplastic strains named A1B1 and 22 protoplastic strains named A2B2, that all were sequenced. The research includes a huge amount of work and the molecular biology of the work seems to be well done. However, the presentation leaves lot to be corrected and requires major revision.

Response: We apologize for the inadequate presentation and have revised the ms clearly and carefully according to your suggestion.

The list of revisions needed, there are probably more

Material and methods

line 93 The G. frondosa dikaryotic bipolar strains LMXY and LMCZ, it is said at the end of Introduction that G. frondose is assumed to be tetrapolar. line 137 The same mistake G. frondosa dikaryotic bipolar strain LMCZ What is meant with bipolar!!

Response: We have changed “dikaryotic bipolar strain” to “dikaryotic strain” (Lines 93, 140)

line 154 Grifola frondosa monokaryotic strain y59 from basidiospores of strain LMCZ/ from a basidiospore of LMCZ.

Response: Grifola frondosa monokaryotic strain y59 from basidiospores of strain LMCZ.

line166 Transfer RNAs were/ Transcript RNAs probably

Response: Agreed and changed (Line 170).

Results

line 263 In the three-round mating experiments (reference? The method It is very well and clearly explained in refrence10, but not here).

Response: Agreed. We have moved it to Materials and Methods (Lines 150-151).

Table 1 is all right, but for Table 2 it is necessary to do a three-round mating experiments with monokaryons grown from individual basidiospores collected from LMCZ fruiting bodies. Then there would be similar distribution of A and B mating -type as from LMXY. Note in Abstract and at the end of the manuscriptit is said Y59 grown from a LMCZ basidospore has the genotype A1B2.

Response: Thanks for pointing that out. In Table 2, we did indeed do three-round mating experiments with monokaryons grown from individual basidiospores collected from LMCZ and LMXY fruiting bodies. For each strain, we also selected three tester strains for three-round mating experiments. LMCZ or LMXY showed the same results, while LMCZ and LMXY were different for the occurrence of pseudo-clamp connections and the limitation of the number of collected basidiospores.

“Y59 grown from a LMCZ basidiospore” has been added to Lines 29-30 and 513

I do not see any sense in the three-round matings between YS-7, YS-5 and YS-11 (protoplastic strain from a dikaryon with mating-types A1B1 and A2B2) and 123 monokaryotic strains grown from single basidiospores of LMCZ. The 123 spores should be used for the three-round experiment suggested above.

Response: There are two mating types, A1B1 and A2B2, in the monokaryotic protoplast strains and four mating types, A1B1, A2B2, A1B2, and A2B1, in the monokaryotic strains from basidiospores. Strains YS-7, YS-5 and YS-11 were from test-cross experiments. YS-7 was an A1B1 type strain, YS-5 and YS-11 were both A2B2 type strains (Line 273). Selecting YS-7, YS-5 and YS-11 with known and different mating types, as tester strains for three-round mating experiments of LMCZ could increase the experimental reliability.

line 287 3.3. Fine genomic map of Grifola frondosa strain LMCZ Y59

Response: Agreed. Y59 has been added (Line 293).

line 299 The fine genomic map of the G. frondosa monokaryotic strain, y59, revealed that the two mating-type loci A and B, were located on chromosomes, 1 and 11. What are the bases of the A and B locus recognization ???

Response: Whether a bipolar or a tetrapolar species, it exists as mating-type A and B loci [9,16,22,25]. The mating-type A locus includes the genes, HD1, HD2, MIP and β-fg, which were determined to be on chromosome 1 of G. frondosa. The mating-type B locus includes genes for the pheromone receptor and precursor and they are located on chromosome 11.

chapter 3.4.1 has to be rewritten, for instance: “There was only one HD2 with two nuclear localization signal sites and one homeodomain in G. frondosa (Figure 2, Supplementary Figure S3B) that were grouped to the HD2s of A. auricula-judae in phylogenetic analysis/ divide in two sentences.

Response: Agreed. It has been changed “There was only one HD2 with two nuclear localization signal sites and one homeodomain in G. frondosa (Figure 2, Supplementary Figure S3B). It was grouped with the HD2s of A. auricula-judae in phylogenetic analysis (Figure 3B, Supplementary Figure S4B)” (Lines 317-320)

For B-receptor and pheromone working in mating the best candidates would be PR3 (seven TMs) and PP3 nice carboxcyl end CVIAA.

Response: Although studies have shown that pheromone receptors such as the PRs of Schizophyllum commune have seven TMs [81], the number of TM domains in PRs does vary among different species. For example, the PRs of Stropharia rugosoannulata contain three to seven TM domains [82] and the PRs of Pleurotus pulmonarius have three to eight TM domains [83]. Just because a PR does not have seven TM domains does not mean that it lacks biological functionality. In G. frondosa, nonsynonymous SNPs and small indel mutations also occur on the HD2, PR1 and PR2 genes, but only PR2 belongs to the PR sublocus; and, all five pheromones have carboxyl ends (CVIAA, Fig. S8). For the B-receptor and pheromone to work together in mating the best candidates would be PR2 and PP2.

The chapters 3.5. and 3.6. dealing with SNP and indels in A and B are poorly written line 363 “We preliminarily inferred that sublocus allelic variation exists on HD2 and PR2 which determines the mating types in G. frondosa, and there may be four different mating types in G. frondosa.” PR2 with only 5 proper TMs?? See the note above

Response: Based on preliminary data, we inferred that the allelic variations existed on subloci of HD2 and PR2, which determined the mating types in G. frondosa and this was tested by reverse genetic experiments. Some species, like Schizophyllum commune have pheromone receptors with seven TM domains [81], but the number varies among different species. For example, the PRs of Stropharia rugosoannulata contain 3-7 TM domains [82], while the PRs of Pleurotus pulmonarius have 3-8 TM domains [83], and both have normal pheromone activity.

Fig.5 text A protoplastic strains A1B1? and B Y59 ?

Response: The mating type of the monokaryotic Y59 strain was A1B2.

Discussion

Segregation distortion? Difficulty to find equal number of the four mating-types???? Is that what the authors mean?

Response: Segregation distortion has been found in many species of edible fungi, such as Agrocybe salicicola [64], L. edodes [65] and Pholiota adiposa [66], and even among some plants [67,68]. Based on our preliminary data, we concluded that the mating type of Grifola frondosa also showed segregation distortion.

line 449-450 …to investigate the dimerization of HD1 and HD2 in future by analyzes of A locus of different monokaryotic strains of LMCZ

Response: Thanks, and changed (Lines 456-457).

line 457 G. lingzhi/ in S11 Ganoderma lucidum

Response: Thanks, and changed (Line 463)

line 467 The number of PRs and their TM domains… The authors should familiarize themselves with the publication What role might non-mating receptors play in Schizophyllum commune. J.of Fungi 7,399 (2021)doi.org/10.33390/jof7050399

Response: Thanks, and added (Line 474)

lines 488-503, what is the purpose of this chapter, could be removed.

Response: In this section, we discuss the problems encountered in performing mating experiments, not only because of the partial separation, but also the inevitable occurrence of pseudo-clamp connections; although, we do have many methods to detect the authenticity of clamp connections. Studies on the A and B mating types and the development of mating gene molecular markers in G. frondosa will be the subject of work in the future.

Supplementary material

text of S4 A ja B impossible to read.

Response: Thanks. Images with higher resolution have been used.

Comments on the Quality of English Language: The English of the text is relatively good, but several sentences could be clearer and thus improve the manuscript.

Response: Thanks for your positive comment. The manuscript has been enhanced.

This manuscript is a resubmission of an earlier submission. The following is a list of the peer review reports and author responses from that submission.

Round 1

Reviewer 1 Report

Comments to Authors

Zhang and colleagues established the theoretical groundwork for precise molecular breeding of G. frondosa, significantly improved hybridization efficiency, and laid the groundwork for developing mating-type molecular markers. The researchers used a wide range of molecular and genome sequencing tools in their investigation. The results of the spore and protoplast isolation, culture, and matting experiments are not detailed in the manuscript. 

Material and Methods

Line number: 94 200 g/L potato to be replaced by 200 g/L potato infusion.

Line 100: PDA is a solid medium, so the word " solid " should be removed. 

Table 3: A few genome data results are unimportant; delete if one is unimportant. 

Figure 2 is unclear, so it will be moved to the supplementary section. If necessary, highlight and enlarge the specific regions. 

Figure 3 is not mandatory, but if possible, include high-resolution images of F. frondosa spores and protoplasts. 

Figures 4 and 5 should have better captions, and Figure 5 needs more resolution.

Author Response

Dear reviewer

Thanks a lot for your suggestions for improving the manuscript. The authors have revised this manuscript follwing your comments. The details are list as belowing.

Responses to Reviewer 1 Comments

Zhang and colleagues established the theoretical groundwork for precise molecular breeding of G. frondosa, significantly improved hybridization efficiency, and laid the groundwork for developing mating-type molecular markers. The researchers used a wide range of molecular and genome sequencing tools in their investigation.

Response: Thanks for the positive comments. We have revised the Ms. carefully according to your suggestions.

The results of the spore and protoplast isolation, culture, and matting experiments are not detailed in the manuscript.

Response: The results of the spore and protoplast isolation have been added (lines 239-241). The results of mating experiments are shown in Tables 1-2 and Supplementary Tables S1-S7.

Line number: 94 200 g/L potato to be replaced by 200 g/L potato infusion.

Response: Agreed and revised (lines 93-94).

Line 100: PDA is a solid medium, so the word " solid " should be removed.

Response: Agreed and removed from line 102.

Table 3: A few genome data results are unimportant; delete if one is unimportant.

Response: Agreed. Table 3 has been deleted.

Figure 2 is unclear, so it will be moved to the supplementary section. If necessary, highlight and enlarge the specific regions.

Response: Agreed. Figure 2 has been enlarged.

Figure 3 is not mandatory, but if possible, include high-resolution images of G. frondosa spores and protoplasts.

Response: Figure 3 has been moved to the supplementary section as Figure S11.

Figures 4 and 5 should have better captions, and Figure 5 needs more resolution.

Response: Agreed and revised. Figures 4 and 5 have been renamed as Figures 3 and 4.

Reviewer 2 Report

These lines of RESULTS move them to DISCUSSION:

Lines 281-283: In tetrapolar ... during mating [57]

Lines 301-302: meaning ... Polyporales [57]

Lines 307-308: The Mat-B ... clamp connections [58]

Lines 317-318: Pheromone receptors ... end inside [59,60]

Lines 332-333: All the PRs ... some edible fungi [18,59,62]

Figure 4: Improve the quality

Author Response

Dear reviewer

Thanks a lot for your suggestions for improving the manuscript. The authors have revised this manuscript follwing your comments. The details are list as belowing.

Responses to Reviewer 2 Comments

These lines of RESULTS move them to DISCUSSION:

Lines 281-283: In tetrapolar ... during mating [57]

Lines 301-302: meaning ... Polyporales [57]

Lines 307-308: The Mat-B ... clamp connections [58]

Lines 317-318: Pheromone receptors ... end inside [59,60]

Lines 332-333: All the PRs ... some edible fungi [18,59,62]

Response: Thanks. Some of the above sentences (lines 301-302, 307-308) have been moved to the discussion section (lines 448-449, 452-453). Some of the sentences (lines 281-283, 317-318) will help explain the findings there. Mat-A and Mat-B, were located on different chromosomes, 1 and 11, respectively (Figure 1). Clearly, G. frondosa is a tetrapolar species (Lines 281-283). Grifola frondosa not only had four PRs (PR1, PR3, PR5, PR6) with seven helical transmembrane-spanning structures, but also PR2 with five TM helices and PR4 with four TM helices based on phylogenetic analysis, amino acid sequence alignment and TMHMM Server results (Lines 317-318). And the sentence (lines 332-333) has been moved to lines 345-346.

Figure 4: Improve the quality

Response: The quality of Figure 4 (renamed Figure 3) has been improved.

Reviewer 3 Report

There are many examples where pheromone and pheromone receptor genes are present in the genome but are not involved in the mating type. The bipolar basidiomycete Pholiota microspora is a typical example. Therefore, not only the gene sequence but also the classical genetic analysis (i.e., presence or absence of clamp cells (phenotype)) and the genotypic linkage should be evaluated to declare that the strain is tetrapolar.

 The LMCZ strain should clearly be evaluated as bipolar, and Strain LMXY should also be evaluated as bipolar.

Author Response

Dear reviewer

Thanks a lot for your suggestions for improving the manuscript. The authors have revised this manuscript follwing your comments. The details are list as belowing.

Responses to Reviewer 3 Comments

There are many examples where pheromone and pheromone receptor genes are present in the genome but are not involved in the mating type. The bipolar basidiomycete Pholiota microspora is a typical example. Therefore, not only the gene sequence but also the classical genetic analysis (i.e., presence or absence of clamp cells (phenotype) and the genotypic linkage should be evaluated to declare that the strain is tetrapolar.

Response: Agree with the reviewer's opinion, some edible fungi do exist pheromone and pheromone receptor genes, but do not have mating type function. This study confirmed that Grifola frondosa belonged to a tetrapolar mating system at both the genetic level and classical genetic analyses. Most SNPs and small indel mutations were distributed on mating loci, but non-synonymous SNPs and small indel mutations also occurred on the HD2, PR1 and PR2 genes; however, only PR2 belonged to the PR sublocus (Figure 1, Supplementary Figure S11). Through a large number of three-round hybridization experiments, four kinds of mating types were obtained of strain LMXY. G. frondosa has four different mating types. Therefore, pheromone receptors and pheromone genes are involved in determining the mating type of G. frondosa.

At the gene level (Result 3.4-3.6), the mating-type A/B loci are located on different chromosomes, and the genes are not linked (lines 291-292); the SNP and indel mutation loci of mating type A and B were identified by comparing the gene sequences of T1, T2 and y59, which further proved that G. frondosa was a tetrapolar heterothallism (lines 355-360). Secondly, classical genetic analysis (three-round mating experiments), confirmed that the mating system of G. frondosa was tetrapolar. Through a large number of three-round hybridization experiments (Result 3.2), it was finally concluded that monokaryotic spores had four mating types in strain LMXY, but there were fewer strains of T3 and T4 mating types due to the existence of pseudo-clamp connection and segregation distortion. Pseudo-clamp connection is also present in Sparassis latifolia and Pleurotus citrinopileatus [10, 85]. Segregation distortion has also been found in many edible fungi species, such as Agrocybe salicacola [63], Lentinula. edodes [64] and Pholiota adiposa [65]. Due to the small number of monokaryotic spore strains obtained and the presence of pseudo-clamp connections, T3 and T4 were not screened out at G. frondosa strain LMCZ; however, combined with genomic data analysis, it was found that there were two other mating strains. Therefore, it was concluded that G. frondosa is a tetrapolar heterothallism.

The LMCZ strain should clearly be evaluated as bipolar, and Strain LMXY should also be evaluated as bipolar.

Response: Strains LMCZ and LMXY can develop fruiting bodies with basidiospores (Supplementary Figure S2A-C), which indicates that each strain is bipolar (added at lines 92-93).

Reviewer 4 Report

In this study, the authors aimed to describe the tetrapolar mating system in the edible mushroom Grifola frondosa. They generated the genome map of G. frondosa and confirmed that it’s a tetrapolar species, so the manuscript has the potential to make an interesting contribution.

However, the manuscript lacks sufficient information to allow the reader to understand what was done and to comprehend the conclusions clearly. The main (but not the only) issues I struggled with are the following:

1.       The process by which protoplasts and basidiospore are obtained is not described.

2.       Please list the fungal strains used in this study.

3.       It is recommended that Figure S2 be placed in the main text. And the microscope photographs are not clear.

How to get protoplasts?

Are protoplasts mononuclear?

4.       The writing style should be uniform. Such as: Mat-A locus, Mate A loci, mating-type loci A and B

5.       Why choose the species in Figure 2?

6.       Line 169, please provide the q-PCR results.

7.       Line 119-120 hard to understand.

8.       Syntax error exists in Line 292, 391, 445…

9.       We cannot find Table in supplementary.

The English language and some terminology lack accuracy.

Round 2

Reviewer 3 Report

The polarity of a mushroom mating type should be determined by its phenotype and not only by the presence or absence of the quadrupolar mushroom mating type gene. In fact, the presence of both A and B genes of tetrapolar mushrooms in bipolar mushrooms has already been reported in many papers.